# Prevalence and Molecular Characteristics of FAdV-4 from Indigenous Chicken Breeds in Yunnan Province, Southwestern China

**DOI:** 10.3390/microorganisms11112631

**Published:** 2023-10-26

**Authors:** Jinyu Lai, Liangyu Yang, Fashun Chen, Xingchen He, Rongjie Zhang, Yong Zhao, Gan Gao, Weiwu Mu, Xi Chen, Shiyu Luo, Tao Ren, Bin Xiang

**Affiliations:** 1College of Veterinary Medicine, Yunnan Agricultural University, Kunming 650201, China; 2College of Veterinary Medicine, South China Agricultural University, Guangzhou 510642, China; 3Center for Poultry Disease Control and Prevention, Yunnan Agricultural University, Kunming 650201, China

**Keywords:** FAdV-4, genetic characterization, indigenous chicken, amino acid mutations

## Abstract

Fowl adenovirus-induced hepatitis–pericardial effusion syndrome outbreaks have been increasingly reported in China since 2015, resulting in substantial economic losses to the poultry industry. The genetic diversity of indigenous chicken results in different immune traits, affecting the evolution of these viruses. Although the molecular epidemiology of fowl adenovirus serotype 4 (FAdV-4) has been well studied in commercial broiler and layer chickens, the prevalence and genetic characteristics of FAdV-4 in indigenous chickens remain largely unknown. In this study, samples were collected from six indigenous chicken breeds in Yunnan province, China. FAdV-positive samples were identified in five of the six indigenous chicken populations via PCR and 10 isolates were obtained. All FAdVs belonged to serotype FAdV-4 and species FAdV-C. The hexon, fiber, and penton gene sequence comparison analysis demonstrated that the prevalence of FAdV-4 isolates in these chickens might have originated from other provinces that exported chicks and poultry products to Yunnan province. Moreover, several distinct amino acid mutations were firstly identified in the major structural proteins. Our findings highlighted the need to decrease inter-regional movements of live poultry to protect indigenous chicken genetic resources and that the immune traits of these indigenous chickens might result in new mutations of FAdV-4 strains.

## 1. Introduction

Fowl adenovirus (FAdV) is a DNA virus belonging to the genus Aviadenovirus in the family Adenoviridae [1]. Based on variations in the antigenic structure, FAdVs can be categorized into three groups: Group I fowl adenoviruses, which are derived from the respiratory tracts and livers of chickens, turkeys, geese, and ducks [2], Group II fowl adenoviruses, primarily isolated from the lymphocytes and livers of turkeys and known as hemorrhagic enteritis viruses (HEV) that primarily induce hemorrhagic enteritis in turkeys and marble spleen disease in pheasants [3], and Group III fowl adenoviruses, referred to as egg drop syndrome viruses (EDSV) [4]. Group I FAdV is a prevalent pathogen in fowl species and can be further divided into five subgroups: A, B, C, D, and E [5]. Additionally, it consists of 12 serotypes, ranging from FAdV-1 to FAdV-11, with FAdV-8 further divided into two types: 8a and 8b [6]. Group I FAdVs causes fowl inclusion body hepatitis (IBH), hydropericardium hepatitis syndrome (HHS), and gizzard erosion (GE), posing significant risks to the poultry sector and impeding its development [7]. Notably, FAdV-4 is the causative agent of HHS, which is distinguished by the presence of pericardial effusion and liver necrosis [8].

The coat proteins of fowl adenoviruses, namely, fiber, penton, and hexon, exhibit a significant degree of homology across all adenoviruses and play a crucial role in determining the efficacy of viral infection [9]. In the case of the group I fowl adenoviruses, the majority possess two fiber proteins, whereas FAdV-4 possesses two fiber proteins of varying lengths [10]. The interactions between viruses and hosts are mediated by fiber proteins [11], whereas viral internalization during the infection cycle is facilitated by penton proteins [12]. Hexon genes, owing to their conserved nature and the antigenicity of the proteins they encode, are utilized as efficient tools for rapid type inference [6].

Chickens of various age groups are susceptible to FAdVs, with broilers aged 3–5 weeks exhibiting the highest vulnerability. Since 2015, there has been a notable increase in reports concerning FAdVs across several provinces in China, particularly broiler-producing provinces, such as Anhui, Shandong, Guangdong, and Guangxi, as evidenced by the substantial number of clinical HHS samples isolated from these regions [13]. Although commercial vaccination against FAdV-4 has been implemented, FAdV-4 infection still occurred sporadically. In most of the previous studies on the isolation and sequencing of FAdVs from commercial broiler and layer chickens [13,14,15], no systematic epidemiological investigation and study of the molecular characteristics of FAdVs from indigenous chickens has been performed.

Indigenous chickens are an important source of genetic and phenotypic diversity. However, infectious diseases threaten the health of chickens. Previous studies have revealed that the immune responses of different kinds of indigenous chicken breeds after being infected with pathogens such as Newcastle disease virus showed significant differences [16,17]. The diversity of the MHC, which regulates specific immune responses, was confirmed in Chinese indigenous chickens [18]. Thus, these viruses might have some mutations under the different selective pressures from these indigenous chickens. Yunnan province, situated in the southwestern region of China, is renowned for its abundant animal resources and is often referred to as the “Kingdom of Animals”. The province has plenty of unique indigenous chicken breeds including the Wuliangshan black-bone chickens, La-bai high leg Chickens, Yanjin black-bone chickens, Xichou black-bone chickens, Yunlong short-foot chickens, Wuding strong chickens, Qinhua chickens, and so on. These indigenous chicken breeds have significant value as precious species resources within China. However, there is a lack of reports on the prevalence of FAdVs in indigenous chickens in this region and the molecular characteristics of these viruses have remained largely unknown. Within the scope of this study, we successfully isolated and identified FAdVs from indigenous chicken populations, thereby elucidating the epidemiological attributes of FAdV-4 in specific chicken populations in Southwest China. These findings suggest that FAdV-4 exists among indigenous chicken breeds in Yunnan province and has some new mutations.

## 2. Materials and Methods

### 2.1. Sample Collection and Processing

To investigate the prevalence of FAdVs in the indigenous chickens in Yunnan province, a total of 478 swab samples including oropharyngeal and cloacal swabs of the same chicken from clinically healthy indigenous chickens and 28 tissue samples from suspected FAdVs infections were collected to detect FAdVs from January 2022 to June 2023. The tissue and swab samples were collected and treated as previously described [19,20]. The samples were obtained by adding tissue to a PBS buffer containing penicillin and streptomycin, followed by grinding in liquid nitrogen, subjecting to three freeze–thaw cycles, and subsequently centrifuging at 4 °C and 8000× *g* to obtain the supernatant. Similarly, the collected swabs underwent three freeze–thaw cycles and were then centrifuged to obtain the supernatant. The supernatant derived from the samples was further filtered through a 0.22 µm filter and stored at a temperature of −80 °C. The specific details of the sample collection are presented in Table 1.

### 2.2. Virus Isolation

The presence of FAdVs was confirmed via polymerase chain reaction (PCR), as described previously [21]. All positive samples were inoculated into LHM cells, according to a previous method [22]. The LMH cells were cultivated in F12 (Gibco, Grand Island, NE, USA) medium supplemented with 10% FBS (Gibco, Grand Island, NE, USA) for the purpose of passaging. Upon reaching 80% confluence, the filtered supernatant derived from the diseased material was diluted by a factor of 5 and subsequently introduced into the LMH cells. Following a 2 hours incubation period, the adsorption process was substituted with F12 medium containing 2% FBS to sustain cell proliferation. The cells were then cultured in 5% CO_2_, 37 °C incubators for a duration of 3 to 5 days. The cells were be monitored daily for signs of swelling, rounding, and aggregation into grape-bunch-like structures. If these characteristics were observed in the majority of cells, the cultured cells and supernatant were collected. After three cycles of freeze–thawing, these samples were filtered using a 0.22 μm filter and then inoculated into LMH cells following the previously described method. After three passages, the cultured cells and supernatant were collected and stored at −80 °C.

### 2.3. DNA Extraction and PCR

Genomic DNA was isolated from the culture supernatants using an Accurate Biology Viral Genomic DNA/RNA Extraction Kit (Accurate Biology, Changsha, China). The samples were analyzed by PCR using specific primers (Appendix A). Analyses were conducted using 1% agarose gel electrophoresis to determine the presence of positive samples. Amplification of the hexon, fiber1, fiber2, and penton genes was performed on these positive samples using the specific primers listed in Appendix A. The target amplification products were purified by 1% agarose gel electrophoresis, and the desired bands were excised. A gel extraction kit (Omega, Norcross, GA, USA) was used to purify the positive products, which were then cloned into a Blunt-zero plasmid vector (Vazyme, Nanjing, China). Subsequently, these clones were transformed into *E. coli* DH5α receptor cells (AlpalifeBio, Shenzhen, China), and the positive clones were selected and sent to Sango Biotech (Shanghai, China) for sequencing.

### 2.4. Sequence Comparison and Phylogenetic Analysis

The nucleotide sequences of hexon, penton, fiber1, and fiber2 were aligned with publicly available reference sequences obtained from the National Library of Medicine (NCBI) using the MegAlign program, which is part of the DNAStar software suite version 5.01 (DNAStar, Madison, WI, USA). Appendix A provides detailed information on the reference sequences. Phylogenetic trees for the hexon, penton, fiber1, and fiber2 genes were constructed using the neighbor-joining method implemented in MEGA version 11.0.

## 3. Results

### 3.1. Virus Identification and Isolation

The chickens suspected to be infected with FAdVs showed obvious lesions on necropsy, with obviously enlarged livers, hemorrhage, and pericardial effusion. The positive rate of swabs and tissue samples were 0.84% (4/478) and 21.43% (6/28), respectively (Table 1). Notably, of the six indigenous chicken flocks in Yunnan province involved in this investigation, we detected the presence of FAdV-4 in five of them. Ten FAdVs were successfully isolated from cultured LMH cells and designated as YNBL-8/2023, YNBL47/2023, YNFD/2022, YNJC/2022, YNLP/2022, YNNC-1/2022, YNNJ/2022, YNXJ/2022, YNBJ-5/2023, and YNYL/2022. The nucleotide sequence information of these FAdVs has been uploaded to NCBI, and the accession number information is shown in Table 2. Notably, four FAdV-4 strains were isolated from the oropharyngeal and cloacal swabs of clinically healthy chickens, indicating an inapparent infection with FAdV-4 in indigenous chicken flocks in Yunnan.

### 3.2. Phylogenetic Analysis of FAdVs Isolated from Indigenous Chickens in Yunnan Province

To investigate the phylogenetic characteristics of FAdVs circulating in indigenous chickens in Yunnan province, the hexon, fiber, and penton genes of the 10 FAdVs were sequenced and submitted to GenBank. Phylogenetic analysis of the hexon, fiber1, fiber2, and penton genes of these 10 FAdVs clustered them into the same branch, which belonged to the FAdV-4/C group (Figure 1, Figure 2 and Figure 3).

The hexon genes of the 10 FAdV-4 strains in this study were compared with the sequences published in the NCBI, and a table of homology between the Yunnan isolates and other isolates was compiled (Appendix A). The hexon genes of all 10 Yunnan isolates were highly homologous to the GDMZ virus isolated from Guangdong province and the SCDY virus isolated from Sichuan province. YNBL-7/2023, YNFD/2022, YNLP/2022, YNNC-1/2022, YNNJ/2022, YNXJ/2022, and YNYL/2022 were also most closely related to AH712 isolated from Anhui province. YNBJ-5/2023, YNBL-7/2023, YNLP/2022, YNNJ/2022, YNXJ/2022, and YNYL/2022 also shared high homology with viruses isolated from Heilongjiang province. YNBL-8/2023 and YNJC/2022 shared 97.72% and 100% identity with GX-1 and GX2017-01, respectively, which were isolated from Guangxi province. High similarity was also observed between YNBL-8/2023, YNFD/2022, and YNYL/2022 and viruses isolated from Shandong province. YNBL-8/2023 and YNXJ/2022 were also the most closely related to the He-Bei/0914/2021 isolate from pigeons.

The fiber1 genes of YNYL/2022, YNXJ/2022, and YNBL-8/2023 were closely related to the isolates from Shandong, Henan, Heilongjiang, and Guangxi. The other strains shared genetic similarities with isolates from the Shandong, Anhui, and Guangdong provinces.

For the fiber 2 gene, YNLP/2022 shared the highest genetic relationship with the Guangxi isolate. The other strains shared genetic similarities with isolates from Shandong, Anhui, and Guangdong provinces in China.

For the penton gene, YNBL-8/2023, YNBL-7/2023, and YNYL/2022 shared the highest genetic relationships with the Henan, Heilongjiang, and Anhui isolates. YNNC-1/2022 and YNXJ/2022 shared the highest genetic relationships with the Shandong isolates. YNJC/2022, YNFD/2022, YNLP/2022, YNBJ-5/2023, and YNNJ/2022 shared the highest genetic relationships with the Henan isolates.

### 3.3. Molecular Characteristics of FAdV-4 Isolated from Southwest China

A thorough examination was undertaken to analyze the amino acid mutations in the sequencing data of the hexon, penton, fiber1, and fiber2 proteins. The investigation revealed the existence of multilocus amino acid mutations, particularly in the hexon protein of the FAdV-4 strain examined in this study. Furthermore, additional amino acid mutations were observed in the penton, fiber1, and fiber2 proteins.

The FAdV-4 hexon protein plays a pivotal role in determining its virulence [23]. Amino acid residue R188 was observed in the hexon proteins of the 10 FAdV-4 isolates, indicating that these viruses are possible highly pathogenic FAdV-4 [24]. Meanwhile, the mutations Y17H in YNBL-7/2023; W97R in YNBL-8/2023; N147D and V816A in YNYL/2022; S164F in YNNJ/2022; R193Q, Q195E, and R290G in YNLP/2022 and YNFD/2022T; 199I, T414I and S429N in YNBL-7/2023; T254S and V914A in YNBL-8/2023; T465A and M707T in YNBJ-5/2023; D588G in YNLP/2022; D588G in YNFD/2022; and G747D in YNXJ/2022 were also identified (Table 3). Notably, the substitutions Q193E and Q195E in YNFD/2022 and YNLP/2022 occurred in the non-pathogenic strain ON1 isolated from Canada and the pathogenic strain MX-SHP95 isolated from Mexico.

The penton base exhibits toxin-like activity that causes cytopathic effects (CPE) in host cells [25]. The penton proteins in our study displayed several mutations, specifically V175E, I193V, T246A, I271V, G402R, V426I, A447T, T486S, T497A, and K509E (Table 4). Notably, the I193V, V426I, and T486S mutations were observed in the avirulent strain ON1 from Canada and the virulent strain MX-SHP95 from Mexico.

The fiber1 protein has been implicated in viral invasion. Prior research has demonstrated that chick embryo lethal orphan (CELO) viruses, specifically FAdV-1, can interact with coxsackieviruses and adenovirus receptors (CARs) through fiber1 when initiating para-infection through fiber 2 proteins [26]. However, the key amino acid sites in fiber1 have not been identified. Compared with ON1 from Canada and MX-SHP95 from Mexico, the isolate from southwest China had a deletion of 428 amino acid sites in fiber1, which was consistent with FAdV-4 isolated from other regions of China. The mutation sites in the fiber1 protein included S25P, T46A, N63S, K165R, T217A, and L255F (Table 5). Among these, aa46 was identical to the strains ON1 from Canada and MX-SHP95 from Mexico.

Previous studies have identified many conserved amino acid mutations in fiber2 proteins, such as G219D, P307A, V319I, and A380T, which are found in all highly virulent FAdV-4 strains, some of which may be associated with virulence [27]. These mutations were also found in isolates from the indigenous chicken breeds of Yunnan province. Five amino acid insertions at aa11–15 was observed in these isolated FAdV-4 strains when compared with the ON1 and MX-SHP95 strains, consistent with the other FAdV-4 strains isolated in China. The fiber2 protein of these viruses also showed some mutations, including E136G, D142G, P384S, S413R, and A475T (Table 6). It is worth noting that the sites D219, Q232, T261, T300, A305, A307, I319, and T380 of the fiber2 protein of theses FAdV-4 strains were identical to the highly virulent MX-SHP95 strain but differed from the ON1 strain.

Amino acid mutations, when combined with breed preference analyses, have revealed that mutations in the hexon protein vary among different chicken breeds. Specifically, strains isolated from Yanjing black-bone and Rongmao chickens harbored R193Q, Q195E, R290G, and D588G mutations. Strains isolated from Wulingshan black-bone chickens harbored the N147D and V816A mutations. Strains isolated from Yanjing black-bone chickens harbored the S164F mutation, which differed between domestic and foreign isolates. Additionally, strains isolated from Xichou black-bone chickens displayed Y17H, W97R, T199I, T254S, S429N, T465A, N601D, W637R, M707T, T735I, and V914A mutations.

## 4. Discussion

FAdVs, especially FAdV-4, can induce high mortality rates in chickens and can be transmitted via horizontal and vertical transmission, posing a great threat to the safety of indigenous chickens. Before 2015, IBH and HHS associated with FAdVs infections, including FAdV-4, FAdV-8a, FAdV-8b, and FAdV-11, were sporadically reported in poultry in China, with low mortality rates [28]. However, since 2015, outbreaks of HHS, mainly associated with FAdV-4, have occurred in broiler-producing provinces such as Shandong, Henan, Jiangsu, Guangdong, Guangxi, and Anhui [29]. Subsequently, the FAdV-4 strain has spread rapidly throughout the country and among layer chickens, resulting in huge economic losses to the poultry industry. FAdV-4 is the dominant serotype, but FAdV-8a, FAdV-8b, and FAdV-11 also are noted in China [22,30]. Indigenous chicken breeds are valuable genetic resources for breeding. Yunnan province, located in southwest China, hosts multiple types of indigenous chickens. In this study, we showed that five of six indigenous chicken populations tested positive for FAdVs. Moreover, all FAdVs detected in this study were identified as FAdV-4. Thus, our results confirmed the prevalence of FAdV-4 but not FAdV-8a, FAdV-8b, or FAdV-11 in indigenous chicken breeds in Yunnan province.

Zhang et al. demonstrated that FAdV-4 strains isolated from central China exhibit clustering with viruses isolated from India, as determined by the hexon sequence, suggesting the potential origin of these viruses as India [31]. However, upon analyzing the genome sequence of three FAdV-4 strains isolated in 2015, it was observed that these strains displayed the highest nucleotide similarity with early Chinese strains. This finding suggests the potential origin of these viruses as early FAdV-4 strains in China [32]. A prior investigation similarly discovered that FAdV-N22 obtained from a live Newcastle disease vaccine, exhibited a substantial degree of sequence similarity with JSJ13, which was isolated from birds affected by infectious bursal disease in China during the period 2012–2013 [33]. This suggests that the contamination of live vaccines may have a significant impact on the occurrence of HHS outbreaks in Chinese poultry. Numerous studies have documented that the viral excretion of FAdV-4 through the respiratory and digestive systems can persist for a duration of two weeks [34,35]. Furthermore, it has been observed that FAdV-4 can be effectively disseminated among fowl populations through aerosol transmission [33]. Recent investigations have identified the presence of FAdV-4 in Taizhou geese [36], Cherry Valley ducks [37], mandarin ducks [38], and Muscovy ducks [38], with a high degree of genetic similarity to viruses isolated from chickens. These findings suggest the potential occurrence of the cross-species transmission of FAdV-4.

The hexon, fiber, and penton gene sequence comparison analysis in this study demonstrated that all 10 FAdV-4 strains isolated from indigenous chickens in Yunnan province had the highest nucleotide similarity with viruses previously reported in other provinces, such as Guangdong, Sichuan, Anhui, Heilongjiang, Shandong, Guangxi, and Henan, indicating that the prevalence of FAdV-4 isolates in these indigenous chickens might have originated from other provinces in China. Although Yunnan province hosts multiple types of indigenous chickens, the productive performance of these indigenous chickens is relatively poorer than that of hybrids of commercial broiler chickens. Thus, most of the chicks were imported into Yunnan province from other provinces, such as Guangdong, Guangxi, Henan, and Shandong. Moreover, poultry products from these developed poultry provinces were brought into Yunnan province to satisfy the consumption needs of the people. This might have resulted in the introduction of FAdV-4 into Yunnan province. It suggests that the scale of broiler breeding in Yunnan province should be expanded to meet the demand of meat and eggs in Yunnan province as much as possible, so as to reduce the demand for live poultry transported from other provinces.

The identification of four essential FAdV-4 genes, specifically hexon, penton, fiber1, and fiber2, has been documented [39,40,41]. Of these genes, hexon and fiber2 are of particular significance in determining the virulence of FAdV-4 and are strongly related to the emergence of highly pathogenic FAdVs [42,43]. The discovery of FAdV-4 in Southwest China implies that FAdV-4 is transmitted between different chicken breeds. The different adaptations of FAdV-4 in indigenous flocks of different breeds have resulted in numerous amino acid mutations in the isolates from Southwest China. Many of the mutations were previously unreported and were concentrated in the hexon and fiber genes. Previous research suggests that the hexon protein is crucial to determining the virulence of FAdV-4. The substitution of the amino acid isoleucine (I) with arginine (R) at position 188 of the hexon protein is anticipated to greatly enhance the virulence of FAdV-4 [44]. In the hexon protein, several mutations have been reported, including I188R, Q193R, and E195Q [45]. However, some mutations, including Y17H, W97R, N147D, T199I, T254S, R290G, N389D, T414I, S429N, T465A, D599G, N601D, W637R, M707T, T735I, G747D, V816A, and V914A, have not been previously reported. Furthermore, the hexon protein from the indigenous chicken isolate displayed mutations at aa193 and aa195, which were identical to those observed in the avirulent strain ON1 from Canada and the virulent strain MX-SHP95 from Mexico. Amino acid mutations also occurred in the fiber1 protein, including the previously unreported mutations S25P, A46T, N63S, K165R, and L255F. In the fiber2 protein, the mutations G219D, P307A, V319I, and A380T were observed in these isolates, which were consistent with previously reported amino acid mutations that may be closely related to virulence [27]. However, the mutations E136G, D142G, E232Q, P384S, S413R, and A475T, which have not been previously reported, were also identified. Notably, the aa46 and aa255 mutations in the fiber1 protein were consistent with strains isolated from other countries and the aa430 mutation was consistent with the virulent Mexican strain MX-SHP95. Previous studies have demonstrated that the MHC system, which plays important roles in immune response, shows high genetic diversity in Chinese indigenous chickens, indicating that the immune traits of these indigenous chickens might have large differences [46]. Recent studies have confirmed that FAdV-4 can induce cellular pathways in chickens to produce interferon and antigen presenting molecules (MHCI/II) [47]. Thus, the different immune traits of these indigenous chickens might have resulted in the mutations in the FAdV-4 strains isolated in our study. However, further research is required to determine the biological significance of these mutations in recombinant chimeric viruses.

In summary, our results reveal the existence of FAdV-4 in indigenous chickens in Yunnan province. Moreover, several mutations in the major structural proteins of FAdV-4 were identified, indicating that FAdV-4 may have undergone evolutionary adaptation to suit the indigenous chicken breeds. Thus, it is imperative to conduct further investigations to ascertain the potential impact of these mutations on viral virulence. More importantly, to protect indigenous chicken breeds in Yunnan province the monitoring and vaccination against FAdVs should be strengthened.

## Figures and Tables

**Figure 1 microorganisms-11-02631-f001:**
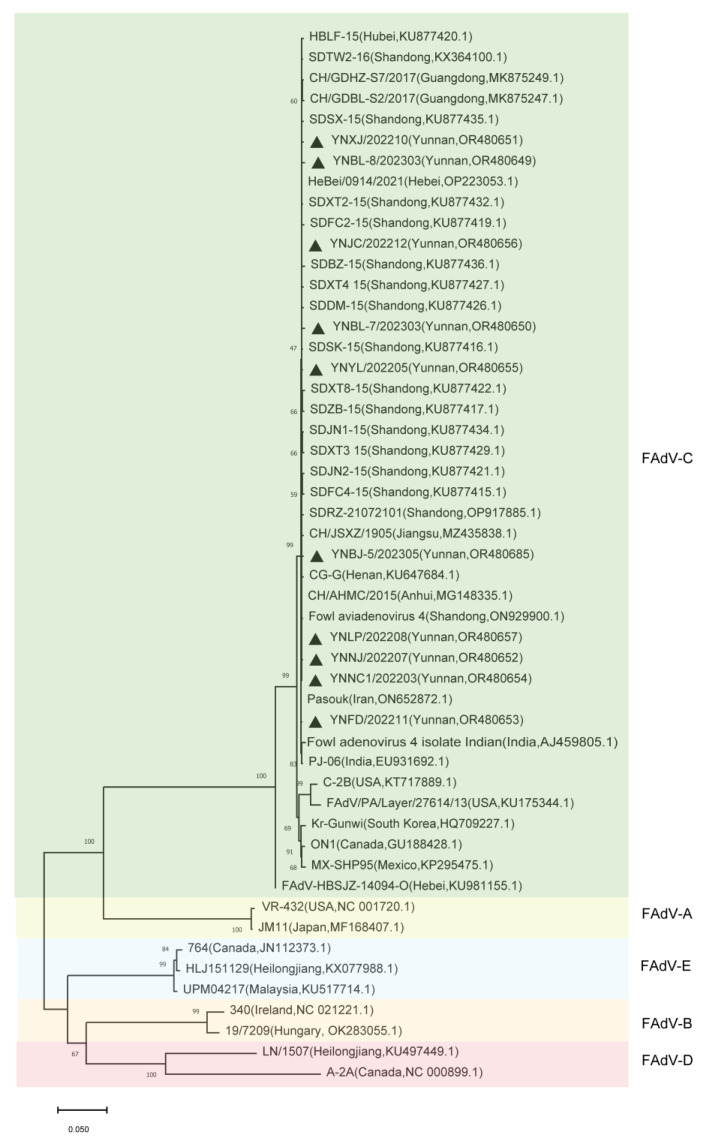
Phylogenetic analysis of the entire hexon gene. Phylogenetic tree for the hexon genes of 10 isolated strains and 41 reference strains created using MEGA 11.0 software using the neighbor-joining method and 1000 bootstrap replicates. The black triangle indicates the isolated strains in this study.

**Figure 2 microorganisms-11-02631-f002:**
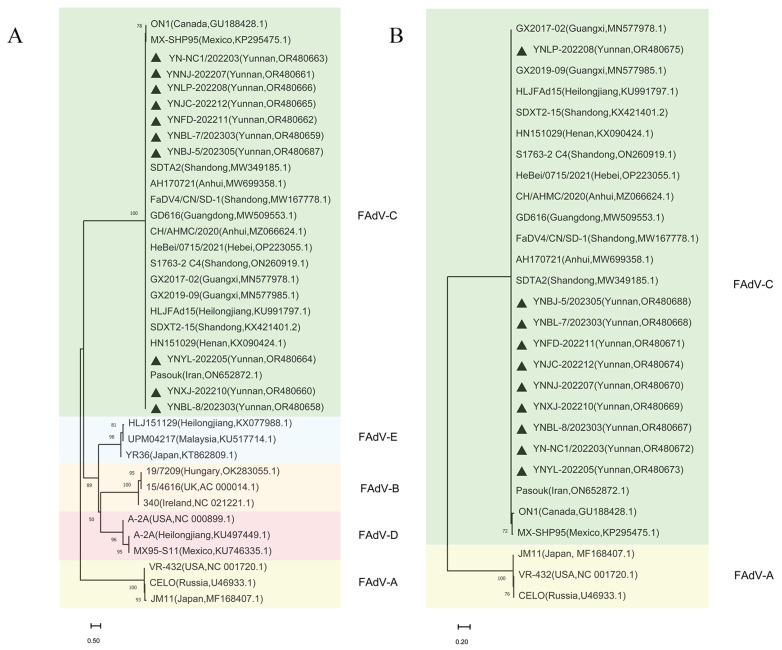
Phylogenetic analysis of the fiber genes. (**A**) Phylogenetic tree for the fiber1 genes of 10 isolated strains and 27 reference strains created using MEGA 11.0 software using the neighbor joining method. (**B**) Phylogenetic tree for fiber2 genes of 10 isolated strains and 18 reference strains created using MEGA 11.0 software using the neighbor-joining method. The black triangle indicates isolated strains in this study.

**Figure 3 microorganisms-11-02631-f003:**
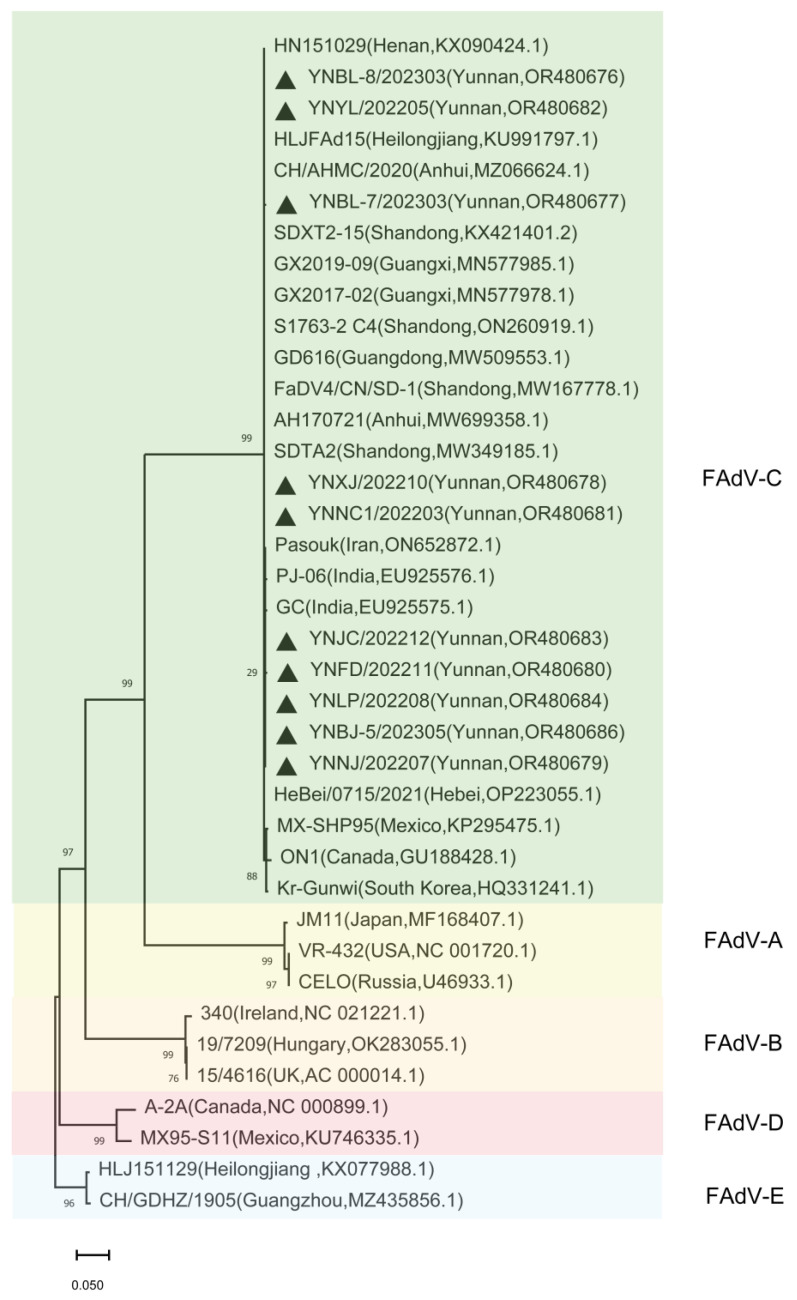
Phylogenetic analysis of the entire penton gene. Phylogenetic tree for penton genes of 10 isolated strains and 28 reference strains created using MEGA 11.0 software using the neighbor-joining method. The black triangle indicates isolated strains in this study.

**Table 1 microorganisms-11-02631-t001:** Surveillance statistics for FAdVs in indigenous chickens in Yunnan province, Southwest China from 2022 to 2023.

Location	Host	Type of Samples	Symptoms	Positive Rate	Isolates	Viral Species
Wenshan	Xichou black-bone chicken	Liver, pericardium, kidneys	Loss of appetite, depression, loose stools	22.22% (4/18)	Xichou black-bone chicken/Yunnan/YNBL-8/2023, Xichou black-bone chicken/Yunnan/YNBJ-5/2023 Xichou black-bone chicken/Yunnan/YNBL-7/2023, Xichou black-bone chicken/Yunnan/YNXJ/2022	FAdV-4
Dali	Wuliangshan black-bone chicken	Mixture of oropharyngeal and cloacal swabs	clinically healthy	0.94% (2/212)	Wuliangshan black-bone chicken/Yunnan/YN-NC1/2022, Wuliangshan black-bone chicken/Yunnan/YNYL/2022	FAdV-4
	Qinhua chicken	Mixture of oropharyngeal and cloacal swabs	clinically healthy	0.78% (1/115)	Qinhua chicken/Yunnan/YNJC/2022	FAdV-4
Nujiang	Lanping rongmao chicken	Mixture of oropharyngeal and cloacal swabs	clinically healthy	3.57% (1/28)	Lanping Rongmao chicken/Yunnan/YNLP/2022	FAdV-4
Lijiang	La-Bai high leg chicken	Mixture of oropharyngeal and cloacal swabs	clinically healthy	0.00% (0/123)	/	
Zhaotong	Yanjing black-bone chicken	Liver, pericardium Liver, pericardium	Loss of appetite, depression and diarrhea	20% (2/10)	Yanjing black-bone chicken/Yunnan/YNNJ/2022, Yanjing black-bone chicken/Yunnan/YNFD/2022	FAdV-4

**Table 2 microorganisms-11-02631-t002:** Accession numbers of all FAdV-4 isolates in this study.

Isolates	Abbreviations	Gene			
		Hexon	Penton	Fiber1	Fiber2
Xichou black-bone chicken/Yunnan/YNBL-8/2023	YNBL-8/2023	OR480649	OR480676	OR480658	OR480667
Xichou black-bone chicken/Yunnan/YNBJ-5/2023	YNBJ-5/2023	OR480685	OR480686	OR480687	OR480688
Xichou black-bone chicken/Yunnan/YNBL-7/2023	YNBL-7/2023	OR480650	OR480677	OR480659	OR480668
Yanjing black-bone chicken/Yunnan/YNFD/2022	YNFD/2022	OR480653	OR480680	OR480662	OR480671
Qinhua chicken/Yunnan/YNJC/2022	YNJC/2022	OR480656	OR480683	OR480665	OR480674
Lanping Rongmao chicken/Yunnan/YNLP/2022	YNLP/2022	OR480657	OR480684	OR480666	OR480675
Wuliangshan black-bone chicken/Yunnan/YN-NC1/2022	YNNC-1/2022	OR480654	OR480681	OR480663	OR480672
Yanjing black-bone chicken/Yunnan/YNNJ/2022	YNNJ/2022	OR480652	OR480679	OR480661	OR480670
Xichou black-bone chicken/Yunnan/YNXJ/2022	YNXJ/2022	OR480651	OR480678	OR480660	OR480669
Wuliangshan black-bone chicken/Yunnan/YNYL/2022	YNYL/2022	OR480655	OR480682	OR480664	OR480673

**Table 3 microorganisms-11-02631-t003:** Amino acid variations in the hexon proteins of the FAdV-4 strains identified in this study.

Isolate	Amino Acids at Position																															
	17	97	147	164	188	193	195	199	238	240	243	254	263	264	290	389	402	410	414	429	465	574	588	601	637	707	735	747	797	816	846	914
YNBJ-5/2023	Y	W	N	S	R	R	Q	T	D	T	N	T	I	V	R	N	A	A	T	S	**A**	I	D	N	W	**T**	**I**	G	P	V	A	V
YNBL-8/2023	Y	**R**	N	S	R	R	Q	T	D	T	N	**S**	I	V	R	**D**	A	A	T	S	T	I	D	N	**R**	M	T	G	P	V	A	**A**
YNBL-7/2023	**H ^a^**	W	N	S	R	R	Q	**I**	D	T	N	T	I	V	R	N	A	A	**I**	**N**	T	I	D	**D**	W	M	T	G	P	V	A	V
YNFD/2022	Y	W	N	S	R	**Q**	**E**	T	D	T	N	T	I	V	**G**	N	A	A	T	S	T	I	**G**	N	W	M	T	G	P	V	A	V
YNLP/2022	Y	W	N	S	R	**Q**	**E**	T	D	T	N	T	I	V	**G**	N	A	A	T	S	T	I	**G**	N	W	M	T	G	P	V	A	V
YNNC-1/2022	Y	W	N	S	R	R	Q	T	D	T	N	T	I	V	R	N	A	A	T	S	T	I	D	N	W	M	T	G	P	V	A	V
YNYL/2022	Y	W	**D**	S	R	R	Q	T	D	T	N	T	I	V	R	N	A	A	T	S	T	I	D	N	W	M	T	G	P	**A**	A	V
YNNJ/2022	Y	W	N	**F**	R	R	Q	T	D	T	N	T	I	V	R	N	A	A	T	S	T	I	D	N	W	M	T	G	P	V	A	V
YNJC/2022	Y	W	N	S	R	R	Q	T	D	T	N	T	I	V	R	N	A	A	T	S	T	I	D	N	W	M	T	G	P	V	A	V
YNXJ/2022	Y	W	N	S	R	R	Q	T	D	T	N	T	I	V	R	N	A	A	T	S	T	I	D	N	W	M	T	**D**	P	V	A	V
PB0505	Y	W	N	S	R	R	Q	T	D	T	N	T	I	V	R	N	A	A	T	S	T	I	D	N	W	M	T	G	P	V	A	V
HLJFAd15	Y	W	N	S	R	R	Q	T	D	T	N	T	I	V	R	N	A	A	T	S	T	I	D	N	W	M	T	G	P	V	A	V
ON1	Y	W	N	T	I	Q	E	T	N	A	E	T	M	I	R	N	A	T	T	S	T	V	D	N	W	M	T	G	A	V	G	V
MX-SHP95	Y	W	N	T	R	Q	E	T	N	A	E	T	M	I	R	N	Q	T	T	S	T	V	D	N	W	M	T	G	A	V	G	V

^a^ Bold-type letters indicate the mutations when compared with the other FAdV-4 strains isolated in this study or ON1 and MX-AHP5.

**Table 4 microorganisms-11-02631-t004:** Amino acid variation in the penton proteins of the FAdV-4 strains identified in this study.

Isolate	Amino Acids at Position													
	42	45	175	193	246	271	356	370	402	426	447	486	497	509
YNBJ-5/2023	P	D	V	I	T	I	V	P	G	**I**	A	**S**	T	K
YNBL-8/2023	P	D	**E** **^a^**	I	T	I	V	P	G	V	**T**	T	T	K
YNBL-7/2023	P	D	V	I	T	I	V	P	**R**	V	A	T	**A**	**E**
YNFD/2022	P	D	V	I	T	**V**	V	P	G	**I**	A	**S**	T	K
YNLP/2022	P	D	V	**V**	T	I	V	P	G	**I**	A	**S**	T	K
YNNC-1/2022	P	D	V	I	T	I	V	P	G	V	A	T	T	K
YNYL/2022	P	D	V	I	**A**	I	V	P	G	V	A	T	T	K
YNNJ/2022	P	D	V	I	T	I	V	P	G	**I**	A	**S**	T	K
YNJC/2022	P	D	V	I	T	I	V	P	G	**I**	A	**S**	T	K
YNXJ/2022	P	D	V	I	T	I	V	P	G	V	A	T	T	K
PB0505	P	D	V	I	T	I	V	P	G	V	A	T	T	K
HLJFAd15	P	D	V	I	T	I	V	P	G	V	A	T	T	K
ON1	S	G	V	V	T	I	A	Q	G	I	A	S	T	K
MX-SHP95	P	G	V	V	T	I	A	Q	G	I	A	S	T	K

^a^ Bold-type letters indicate the mutations when compared with the other FAdV-4 strains isolated in this study or ON1 and MX-AHP5.

**Table 5 microorganisms-11-02631-t005:** Amino acid variation in the fiber1 proteins of the FAdV-4 strains identified in this study.

Isolate	Amino Acids at Position																					
	14	25	28	44	46	63	69	70	119	126	153	165	186	196	204	217	251	255	262	263	428	430
YNBJ-5/2023	A	S	S	R	T	N	G	S	N	A	R	K	D	V	G	T	L	L	H	D	-	G
YNBL-8/2023	A	S	S	R	**A**	**S**	G	S	N	A	R	**R**	D	V	G	T	L	**F**	H	D	-	G
YNBL-7/2023	A	S	S	R	**A**	N	G	S	N	A	R	K	D	V	G	T	L	L	H	D	-	G
YNFD/2022	A	S	S	R	**A**	N	G	S	N	A	R	K	D	V	G	T	L	**F**	H	D	-	G
YNLP/2022	A	S	S	R	**A**	N	G	S	N	A	R	K	D	V	G	T	L	**F**	H	D	-	G
YNNC-1/2022	A	S	S	R	**A**	N	G	S	N	A	R	K	D	V	G	T	L	**F**	H	D	-	G
YNYL/2022	A	**P ^a^**	S	R	**A**	N	G	S	N	A	R	K	D	V	G	T	L	**F**	H	D	-	G
YNNJ/2022	A	S	S	R	**A**	N	G	S	N	A	R	K	D	V	G	T	L	L	H	D	-	G
YNJC/2022	A	S	S	R	**A**	N	G	S	N	A	R	K	D	V	G	T	L	L	H	D	-	G
YNXJ/2022	A	S	S	R	**A**	N	G	S	N	A	R	K	D	V	G	**A**	L	**F**	H	D	-	G
PB0505	A	S	S	R	T	N	G	S	N	A	R	K	D	V	G	T	L	L	H	D	-	G
HLJFAd15	A	S	S	R	T	N	G	S	N	A	R	K	D	V	G	T	L	L	H	D	-	G
ON1	V	S	I	P	A	N	S	G	D	V	H	K	N	T	G	T	I	L	Q	E	H	S
MX-SHP95	V	S	I	P	A	N	S	G	D	V	H	K	N	V	A	T	I	L	Q	E	N	G

^a^ Bold-type letters indicate the mutations when compared with the other FAdV-4 strains isolated in this study or ON1 and MX-AHP5.

**Table 6 microorganisms-11-02631-t006:** Amino acid variation in the fiber2 proteins of the FAdV-4 strains identified in this study.

Isolate	Amino Acids at Position																	
	11–15	22	29	114	136	142	144	219	232	261	300	305–307	319	380	384	413	475	478
YNBJ-5/2023	ENGKP	S	A	D	E	D	S	D	Q	T	T	ANA	I	T	P	S	A	L
YNBL-8/2023	ENGKP	S	A	D	E	D	S	D	Q	T	T	ANA	I	T	P	S	A	L
YNBL-7/2023	ENGKP	S	A	D	E	D	S	D	Q	T	T	ANA	I	T	P	S	A	L
YNFD/2022	ENGKP	S	A	D	E	D	S	D	Q	T	T	ANA	I	T	P	S	A	L
YNLP/2022	ENGKP	S	A	D	E	D	S	D	Q	T	T	ANA	I	T	P	S	A	L
YNNC-1/2022	ENGKP	S	A	D	E	D	S	D	Q	T	T	ANA	I	T	P	**R**	**T**	L
YNYL/2022	ENGKP	S	A	D	**G ^a^**	**G**	S	D	Q	T	T	ANA	I	T	**S**	S	A	L
YNNJ/2022	ENGKP	S	A	D	E	D	S	D	Q	T	T	ANA	I	T	P	S	A	L
YNJC/2022	ENGKP	S	A	D	E	D	S	D	Q	T	T	ANA	I	T	P	S	A	L
YNXJ/2022	ENGKP	S	A	D	E	D	S	D	Q	T	T	ANA	I	T	P	S	A	L
PB0505	ENGKP	S	A	D	E	D	S	D	Q	T	T	ANA	I	T	P	S	A	L
HLJFAd15	ENGKP	S	A	D	E	D	S	D	Q	T	T	ANA	I	T	P	S	A	L
ON1	-	S	P	D	E	D	S	G	E	S	I	SHP	V	A	P	T	A	V
MX-SHP95	-	Y	P	A	E	D	A	D	Q	N	T	AHA	I	T	P	T	A	V

^a^ Bold-type letters indicate the mutations when compared with the other FAdV-4 strains isolated in this study or ON1 and MX-AHP5.

## Data Availability

The genomic data presented in this study are available from GenBank (accession numbers: OR480649-OR480688).

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
