# Peer review of "Prevalence and Molecular Characteristics of FAdV-4 from Indigenous Chicken Breeds in Yunnan Province, Southwestern China"

_microorganisms, 2023, doi:10.3390/microorganisms11112631_

Round 1

Reviewer 1 Report

Dear Authors

In my opinion, your manuscript needs improvement. Please see the comments.

line 19 - "In this study, samples were collected from six local chickens in Yunnan Province, China" - do you mean local breeds? It seems like you were sampling just 6 birds

 line 20 "FAdV-positive samples were identified in five of six in digenous chicken populations via PCR and 10 FAdVs were successfully isolated" - rearrange the sentence - I suggest "...via PCV and 10 isolates were obtained"

line 69 - are in Yunnan province the conservative flocks (the breeding flocks  with protection of their genetic resources)?  The line 26 in abstract suggests it, but it is not clear. Are those breeds present in other parts of China?

line 85 - were the 478 swabs from both pharynx and cloaca (mixed) - which gives 478 birds, or half from pharynx and half from cloaca - 238 examined  birds? The table 1 do not clarify it

line 141 - "five of the six indigenous chicken populations were FAdV-positive" rearrange the sentence if because you examined just a group of birds - if it is not the only reproductive flock in China, you could not write about the population

figures 1,2,3 - it would be good to add the provinces next to the isolates number, to show the relatedness the isolates according the regions. The table 3 could be as a supplementary file then

line 270 - change word "exist" to "are noted"

lines 306-310 - is it possible?

line 344 - " you can's write : "these viruses might have originated from other provinces in China" when in introduction you're writing that " there is a lack of reports on the prevalence of FAdV in the local chicken in this region" - so the previous status was unknown

Some editorial errors exists, f.e.

line 52- Hexon (capital letter is missing)

or space lacks (f.e.39, 101, 108, 278, 285, 289)

or 2 spaces are present (f.e. 89, 144, 277, 280, 292)

Author Response

Dear Reviewer,

  We express our gratitude for your invaluable advice on our manuscript microorganisms-2647369, which has greatly contributed to our work. In the revised manuscript, we have highlighted all the modifications in yellow, according to your suggestions.

  1. line 19 - "In this study, samples were collected from six local chickens in Yunnan Province, China" - do you mean local breeds? It seems like you were sampling just 6 birds

Reply: An error was identified in this section of the description. In this study, samples were collected from six indigenous chicken breeds in Yunnan Province, China. This error has been rectified and duly noted in line 21.

  1. line 20 "FAdV-positive samples were identified in five of six in digenous chicken populations via PCR and 10 FAdVs were successfully isolated" - rearrange the sentence - I suggest "...via PCV and 10 isolates were obtained"

Reply: We've revised the sentence based on your comments(line 22).

3.line 69 - are in Yunnan province the conservative flocks (the breeding flocks  with protection of their genetic resources)?  The line 26 in abstract suggests it, but it is not clear. Are those breeds present in other parts of China?

Reply: Thanks for your carefully reading and nice advice. The samples were collected from indigenous chicken breeds in Yunnan Province, where these breeds are protected, and these breeds are endemic to Yunnan Province and not found elsewhere(lines 74~77). 

4.line 85 - were the 478 swabs from both pharynx and cloaca (mixed) - which gives 478 birds, or half from pharynx and half from cloaca - 238 examined  birds? The table 1 do not clarify it

Reply: Thanks for your carefully reading and nice advice. Oropharyngeal and cloacal swabs from the same chicken were mixed during sampling.We have corrected it in Table 1and Line 90.

5.line 141 - "five of the six indigenous chicken populations were FAdV-positive" rearrange the sentence if because you examined just a group of birds - if it is not the only reproductive flock in China, you could not write about the population

Reply:  Thanks for your carefully reading and nice advice. We apologise for the error in the language description and have amended this part of the description(line 145-146).

  1. figures 1,2,3 - it would be good to add the provinces next to the isolates number, to show the relatedness the isolates according the regions. The table 3 could be as a supplementary file then

Reply:  Thanks for your advice, and we completely agree the reviewer’s suggestion. We have modified figures 1,2 and 3 in response to your comments by adding the  the information of province after the sequence name. Moreover, Table 3 will be made available as a supplementary document.

  1. line 270 - change word "exist" to "are noted

Reply:Thanks for your advice, and we completely agree the reviewer’s suggestion.  Changes have been made in response to the comments you have made(line 266)

  1. lines 306-310 - is it possible?

Reply: :Thanks for your advice. According to statistics from the Yunnan Poultry Association, Yunnan Province relies on inputs from the other province for most of its commercial chickens, but the data provided in our manuscript are not up-to-date, so we have made a correction on lines 298~301.

  1. line 344 - " you can'twrite : "these viruses might have originated from other provinces in China" when in introduction you're writing that " there is a lack of reports on the prevalence of FAdV in the local chicken in this region" - so the previous status was unknown

Reply: Thanks for your advice, and we completely agree the reviewer’s suggestion. According to your comments, it has been modified in lines 338-339.

10 .Some editorial errors exists, f.e.

line 52- Hexon (capital letter is missing)

or space lacks (f.e.39, 101, 108, 278, 285, 289)

or 2 spaces are present (f.e. 89, 144, 277, 280, 292)

Reply: Thanks for your advice. According to your comments, it has been modified.

Thank you for your valuable comments.

Wish you have a nice day!

Sincerely,

Bin,Xiang

Reviewer 2 Report

1)    Title : « local chickens » worth to be replaced by « indigenous chicken breeds » 

2)    Introduction: could you add any information related to FAdV prevention used

3)    Line 19 : samples were collected from « six local chickens » should be replaced by « collected from six local chicken breeds »

4)    Line 87 : Could you add the range of chicken age concerned by this study

5)    Line 88 : more information needed such as samples were collected from sick or dead birds , more information on clinical cases  from where samples were collected (age,  symptoms, mortality rates..)

6)    Line 86: precise if the oropharyngeal and cloacal swabs were pooled or investigated separately. 

7)    Line 146: how many positive samples from oropharyngeal and from cloacal  in case of swabs were examined separately? 

Author Response

Dear Reviewer,

  Thank you very much for your valuable comments on our manuscript microorganisms-2647369. In accordance with your suggestions, we have implemented the following modifications to the manuscript. In the revised manuscript, we have highlighted all the modifications in yellow,according to your suggestions.

1)    Title : « local chickens » worth to be replaced by « indigenous chicken breeds » 

Reply: Thank you for your good advice. We've revised it in the title and somewhere else (line 3)

2) Introduction: could you add any information related to FAdV prevention used

Reply: Thank you for your good advice. We've added this to lines 60~62.

3) Line 19 : samples were collected from « six local chickens » should be replaced by « collected from six local chicken breeds »

Reply: Thank you for your good advice. We've revised the sentence based on your comments(line 21).

4)  Line 87 : Could you add the range of chicken age concerned by this study.

Reply: Thanks for your carefully reading and nice advice. A previous studied have systematically investigated the effect of age on the susceptibility of chickens to  FAdV-4 (Yuan et al. ,2021). In our study, we focused on investigation of   the prevalence and genetic characteristics of FAdV-4 in local chickens from Yunnan province. Therefore, we were sorry that we did not collected the age of these sampled chickens. We’ll avoid similar problems in the future.

Yuan F, Song H, Hou L, Wei L, Zhu S, Quan R, Wang J, Wang D, Jiang H, Liu H, Liu J. Age-dependence of hypervirulent fowl adenovirus type 4 pathogenicity in specific-pathogen-free chickens. Poult Sci. 2021 Aug;100(8):101238. doi: 10.1016/j.psj.2021.101238.

5) Line 88 : more information needed such as samples were collected from sick or dead birds , more information on clinical cases  from where samples were collected (age,  symptoms, mortality rates..)

Reply: Thank you for your good advice. We have added information on symptoms that were collected in Table 1 and on autopsy symptoms in lines 142 through 143. Age information was not collected for objective reasons.

6) Line 86: precise if the oropharyngeal and cloacal swabs were pooled or investigated separately. 

Reply: Thanks for your carefully reading. Oropharyngeal and cloacal swabs from the same chicken were mixed during sampling. We have corrected this point in Table 1and line 90.

7)Line 146: how many positive samples from oropharyngeal and from cloacal  in case of swabs were examined separately? 

Reply: We tested oropharyngeal and cloacal swabs from 478 chickens, and the two types of swabs from the same chicken were combined. Among these swabs, a total of four FAdVs positive samples were detected, yielding an isolation rate of 0.84%. After LMH cell isolation, a total of 4 FAdV-4 strains were isolated.The results are described in detail in lines 143-144.

Thank you for your valuable comments.

Wish you have a nice day!

Sincerely,

Bin,Xiang